# Randomised Controlled Trial: Partial Hydrolysation of Casein Protein in Milk Decreases Gastrointestinal Symptoms in Subjects with Functional Gastrointestinal Disorders

**DOI:** 10.3390/nu12072140

**Published:** 2020-07-18

**Authors:** Reijo Laatikainen, Hanne Salmenkari, Timo Sibakov, Heikki Vapaatalo, Anu Turpeinen

**Affiliations:** 1Booston Oy Ltd., Viikinkaari 6, FI-00790 Helsinki, Finland; 2Pharmacology, Medical Faculty, University of Helsinki, P.O. Box 63, FI-00014 Helsinki, Finland; hanne.salmenkari@helsinki.fi (H.S.); heikki.vapaatalo@helsinki.fi (H.V.); 3Valio Ltd., R&D, P.O. Box 30, FI-00039 Valio, Finland; timo.sibakov@valio.fi (T.S.); anu.turpeinen@valio.fi (A.T.)

**Keywords:** IBS, bloating, functional gastrointestinal disorder, milk, casein, hydrolysation, inflammation

## Abstract

Unspecific gastrointestinal symptoms associated with milk consumption are common. In addition to lactose, also other components of milk may be involved. We studied whether the partial hydrolysation of milk proteins would affect gastrointestinal symptoms in subjects with functional gastrointestinal disorders. In a randomised, placebo-controlled crossover intervention, subjects (*n* = 41) were given ordinary or hydrolysed high-protein, lactose-free milkshakes (500 mL, 50 g protein) to be consumed daily for ten days. After a washout period of ten days, the other product was consumed for another ten days. Gastrointestinal symptoms were recorded daily during the study periods, and a validated irritable bowel syndrome-symptom severity scale (IBS-SSS) questionnaire was completed at the beginning of the study and at the end of both study periods. Blood and urine samples were analysed for markers of inflammation, intestinal permeability and immune activation. Both the IBS-SSS score (*p* = 0.001) and total symptom score reported daily (*p* = 0.002) were significantly reduced when participants consumed the hydrolysed product. Less bloating was reported during both study periods when compared with the baseline (*p* < 0.01 for both groups). Flatulence (*p* = 0.01) and heartburn (*p* = 0.03) decreased when consuming the hydrolysed product but not when drinking the control product. No significant differences in the levels of inflammatory markers (tumor necrosis factor alpha, TNF-α and interleukin 6, IL-6), intestinal permeability (fatty acid binding protein 2, FABP2) or immune activation (1-methylhistamine) were detected between the treatment periods. The results suggest that the partial hydrolysation of milk proteins (mainly casein) reduces subjective symptoms to some extent in subjects with functional gastrointestinal disorders. The mechanism remains to be resolved.

## 1. Introduction

Functional gastrointestinal disorders (FGDs) are common and often difficult-to-treat ailments [1]. FGDs include both disorders of the lower gastrointestinal tract (GI), such as irritable bowel syndrome (IBS), and disorders of the upper GI tract, such as functional dyspepsia [2]. In addition to IBS and dyspepsia, other rather common FGDs include functional bloating, functional constipation and functional diarrhoea [2]. Rome IV criteria are used in the differential diagnosis of these and other of FGDs in clinical trials [2], but they have been less often used in clinical practice.

The treatment of FGDs remains challenging; the efficacy of different drug treatments is often suboptimal, and adverse effects of many previously marketed drugs have led to market withdrawals [1]. People with FGDs commonly report intolerances to particular food items, such as milk, wheat, onions, garlic, chili, beans and coffee [3,4]. During the last decade, a low FODMAP (fermentable oligo-, di-, monosaccharides and polyols) diet has attracted attention and gained acceptance as a dietary means of treating IBS patients with food intolerance, but it is rarely the ultimate solution [5]; many patients experience residual symptoms, and not all individuals respond to a low FODMAP diet [5].

Milk and other lactose-containing dairy products may cause functional gastrointestinal symptoms among people with hypolactasia. It is of interest that 40% of 1900 Finnish subjects with unspecific GI symptoms visiting an outpatient clinic reported that their GI symptoms were milk-related [6]. However, the prevalence of hypolactasia in the Finnish population is only around 17% [7]. Therefore, there may well be additional factors in milk other than lactose that play a role in triggering GI symptoms. The role of milk proteins in functional disorders has been a less extensively investigated subject. Some preliminary interventions in diverse patient groups have shown that the milk protein, casein or milk, per se, with intact proteins might induce gastrointestinal symptoms in some individuals [8,9,10]. Therefore, the hydrolysis of milk proteins, in addition to lactose, might be one way to improve the tolerability of dairy products. Indeed, the first randomised study scrutinising the effect of casein-hydrolysed milk revealed an enhanced tolerability to milk with hydrolysed β-casein when compared to milk with intact protein; a difference between the milk products was found in flatulence, rumbling symptoms and the overall symptoms score favouring β-casein hydrolysed milk [11]. These findings underline the need for further studies in order to understand the specific role of milk proteins in the treatment of FGDs. We investigated the hypothesis that gastrointestinal symptoms in patients with FGDs could be reduced by hydrolysation of β-casein into smaller peptides.

## 2. Materials and Methods

### 2.1. Subjects and Study Design

Subjects were recruited via advertisements in Facebook and the internet (www.pronutritionis.net and www.tervevatsa.fi). The main inclusion criteria were IBS, functional dyspepsia, functional diarrhoea or functional bloating, according to Rome IV criteria. Further inclusion criteria included the ability to consume 500 mL lactose-free milk products daily during the intervention and ages between 18 and 65 years. Subjects were excluded if they had a diagnosis of inflammatory bowel disease, milk allergy or cancer. In addition, pregnant and lactating women, alcoholics or subjects taking medications potentially influencing their gastrointestinal functions and subjects participating in any other clinical trials were not eligible.

A total of 41 subjects, 33 women and 8 men, were recruited. The subjects had been given a diagnosis of IBS (*n* = 23), functional dyspepsia (*n* = 3), functional diarrhoea (*n* = 6) or functional bloating (*n* = 9). Prescreening of candidates was done in a telephone interview. Subjects meeting preliminary inclusion criteria were invited to a screening study visit, where their health status, possible medications and dietary restrictions were evaluated. Before entering the study, all subjects provided written informed consent.

Baseline characteristics of the subjects are presented in Table 1.

The study was a randomised, placebo-controlled, crossover intervention of two groups with ten-day treatment periods, separated by a washout period of ten days (Figure 1). Subjects were randomised to the two groups in blocks of four. A person who was not involved with enrolling the participants or assigning them to the interventions generated the randomisation list. Both the investigators and the subjects were blinded to the randomisation.

The subjects were instructed to maintain their habitual diets and the usual level of physical activity throughout the study. Illnesses, medications or other deviations from normal were recorded daily in their electronic study diaries. The intake of foods containing dairy proteins was limited during the ten-day study periods, and the subjects were given a list of such food products. The subjects kept food records for three days at the end of both treatment periods, and macronutrient and fibre intakes were calculated for both periods.

The study was approved by the Ethics Committee of the Hospital District of Helsinki and Uusimaa (HUS/576/2019). The study was registered at ISRCTN registry (https://www.isrctn.com) with identification number ISRCTN74158117.

### 2.2. Study Products and Diet

Chocolate-flavoured, lactose-free, high-protein milkshakes were produced by Valio Ltd, Helsinki, Finland. The control milkshake was a commercially available product, whereas the hydrolysed product was prepared specially for this study.

The lactose-free milkshake was produced from ultra-filtered milk concentrate. During ultrafiltration, a part of the lactose was removed, and proteins were concentrated. After ultrafiltration, the residual lactose and part of the proteins were hydrolysed by enzymatic hydrolysis. The lactose content of the milkshakes was <0.01%. Protein hydrolysis was undertaken according to patent EP 2632277B1 [12]. The protein content of the milkshakes was 10%. The hydrolysis of the proteins was controlled such that the degree of hydrolysis was 6.3 mg free tyrosine/g protein in the test milkshake, as analysed according to the modified method of Matsubara et al. [13]. The samples were boiled for 4 min at 100 °C and centrifuged prior to analysis. Soluble tyrosine was determined from the supernatant after centrifugation (3000 rcf, 15 min). According to the results of the reversed-phase high-performance liquid chromatography method [14], the concentration of β-casein was reduced by 75%, as compared to the concentration in the control milkshake.

According to sodium dodecyl sulphate-polyacrylamide gel electrophoresis (SDS-PAGE), whey proteins were not hydrolysed significantly [15] (Figure 2). SDS-PAGE analyses were carried out using ready-made 12% Tris-glycine extended polyacrylamide gels (Bio-Rad, Hercules, CA, USA). The amount of protein added to each sample well was 10 μg. Protein bands were stained with Coomassie Brilliant Blue R-250 (Bio-Rad, Watford, UK) and compared with molecular weight markers Precision Plus Protein All Blue Standards (Bio-Rad Laboratories, Hercules, CA, USA).

During the ten-day study periods, each day, the subjects consumed two bottles of the products (i.e., 250 mL twice daily), which provided 50 g of milk protein and about 300 kJ of energy. The nutritional composition of the study products is shown in Table 2. Products were packed in similar unlabelled cartons and were differentiated with a code on the cap. Subjects were allowed to decide at what time of the day they consumed the products. The time of day and amount consumed were recorded in an electronic study diary.

The dietician instructed the subjects to follow an otherwise dairy-free diet during the ten-day study periods. The only dairy products permitted were the study milkshakes. Milk, sour milk, kefir, yogurt, quark, cream, cheese or other dairy products were to be excluded, but plant-based alternatives, such as oat milk and vegan cheese, were allowed; adherence to these restrictions was monitored with food diaries.

### 2.3. Gastrointestinal Symptoms

The occurrence of gastrointestinal symptoms—abdominal pain, flatulence, rumbling, bloating, diarrhoea, constipation, heartburn and rapid feeling of fullness—was recorded in an electronic study diary, which the subjects completed for three days before the first study period (baseline, subjects’ habitual diet) and daily during both ten-day study periods. The severity of the subjective symptoms was evaluated on a scale from 1 (no discomfort) to 5 (severe discomfort). The total symptom score was then calculated by summing scores for all eight individual symptoms (max score 40).

The irritable bowel syndrome-symptom severity scale (IBS-SSS) questionnaire was filled at the baseline before the beginning of the study and at the end of both study periods. This questionnaire evaluated the gastrointestinal pain, bowel dysfunction, IBS severity and overall wellbeing, with scores being based on a visual analogue scale [16].

### 2.4. Urine Samples

At the beginning of the study and at the end of both study periods, a 12-h (overnight) urine sample was collected. Urine was collected into a container containing 7 mL 6 N hydrochloric acid as the preservative. During collection, the container was stored at 4 °C, and subjects returned it to the study centre on the same day that the collection was finished. The volume of the urine was measured, and after aliquoting, samples were stored at −70 °C. Urine was analysed for a marker of immune activation, 1-methylhistamine. The concentration of 1-methylhistamine was adjusted for the 12-h urinary creatinine excretion.

### 2.5. Blood Samples

Fasting blood samples (20 mL) were collected at the beginning of the study and at the end of both study periods from the antecubital vein. Plasma was separated, and the samples were stored at −70 °C. Plasma was analysed for two markers of inflammation (interleukin -6, IL-6 and tumor necrosis factor alpha, TNF-α) and one of intestinal permeability (fatty acid binding protein 2, FABP2).

### 2.6. Laboratory Methods

The proinflammatory cytokines, TNF-α (HSTA00E) and IL-6 (HS600C), and the intestine-derived FABP2 (DY3078) were measured from the plasma samples using ELISA kits (R&D Systems, Minnesota, MO, USA). 1-Methylhistamine (DLD Diagnostika GmbH, Hamburg, Germany) and creatinine (Invitrogen, Thermo Fisher Scientific, Inc., Waltham, MA, USA) were analysed from urine; the values were calculated as ng 1-methylhistamine per mg of creatinine.

### 2.7. Statistical Analyses

The sample size calculation was based on presumable changes in the IBS-SSS. Suitable previously published data were not available to be used in the power calculations. Therefore, we assumed that the difference between the study products would be at least 50 points on the 500-point IBS-SSS score and that the standard deviation of that difference would be 100 points. Thus, a sample size of 34 would have 80% power to detect this 50-point difference when using a paired *t*-test with a 0.05 two-sided significance level. The anticipated drop out was 10–20%, and therefore, 40–44 patients were targeted for this crossover study.

Daily reported gastrointestinal symptoms (abdominal pain, flatulence, rumbling, bloating, diarrhoea, constipation, heartburn and rapid feeling of fullness); the total daily symptom score and the IBS-SSS total score were the primary outcome variables. Symptom scores for all individual symptoms were calculated for the baseline (mean of three days) and for the study periods (mean of ten days). A total daily symptom score was calculated for the gastrointestinal symptoms (mean of eight symptoms). A repeated-measures analysis of variance (ANOVA) for a crossover design was used to compare individual symptoms, total symptom scores and IBS-SSS scores between the study periods. Comparisons to baseline were done as post hoc analyses.

Secondary outcome variables included two inflammatory markers (plasma IL-6 and TNF-α), one marker of intestinal permeability (plasma FABP2) and a marker of immune activation (urinary 1-methylhistamine). The results for the secondary outcomes were compared using repeated measures analysis of variance (ANOVA) for the crossover design. The distribution of TNF-α was skewed, and therefore, data was logarithmically transformed before analysis.

Background diets during both study periods were monitored by using three-day food diaries. Dietary intakes of macronutrients and fibre were calculated by using the Fineli database available at its internet interface at www.fineli.fi. Paired *t*-test was used to analyse differences between the study periods. The statistical analyses were performed using IBM SPSS Statistics for Windows (version 26.0, IBM Corp, Armonk, NY, USA). A *p*-value < 0.05 was considered statistically significant.

## 3. Results

### 3.1. Compliance

Thirty-seven subjects completed the study. Two subjects dropped out due to personal reasons and two subjects due to unpleasant gastrointestinal symptoms. The flow chart is depicted in the Appendix A.

Based on study diaries, where the subjects recorded the consumption of study products daily, compliance with the study products was excellent. On average, 490 mL/day (of 500 mL/day) of the control product and 480 mL/day (of 500 mL/day) of the hydrolysed product were consumed. The taste of both products was assessed as good.

### 3.2. Gastrointestinal Symptoms

The IBS-SSS total symptom score was lower during the hydrolysed period than during the control period (*p* = 0.001). The score also decreased significantly from the baseline during the period with the hydrolysed product (*p* = 0.001) but not during the period when they drank the control milkshake (*p* = 0.68). The total daily symptom score was significantly lower after consuming the hydrolysed product in comparison with both the control product (*p* = 0.002) and the baseline symptom score (*p* = 0.001, Table 3).

Less heartburn was reported during the hydrolysed period (*p* = 0.01), while no significant differences in abdominal pain, flatulence, rumbling, bloating, diarrhoea, constipation or feeling of fullness were seen between the treatment periods (*p* > 0.05, Table 3). Significantly less bloating was reported during both study periods when compared with the baseline (*p* < 0.01 for both groups). Flatulence (*p* = 0.01) and heartburn (*p* = 0.03) decreased during the consumption of the hydrolysed product when compared to the baseline but not during the period when they drank the control milkshake.

### 3.3. Inflammatory Markers, Intestinal Permeability and Immune Activation

Plasma IL-6 concentrations increased during the hydrolysed period (*p* = 0.02) but did not differ significantly from the control period (*p* = 0.36, Table 4). No significant changes were seen in plasma FABP2 or TNF-α concentrations between either of the treatment periods or as compared to the baseline.

The level of urinary 1-methylhistamine decreased significantly from the baseline during the period when they drank the hydrolysed milkshake (*p* = 0.03), but no difference was evident between the study periods (*p* = 0.43).

### 3.4. Dietary Intakes

Dietary intakes during the treatment periods are reported in Table 5. The median intake of the energy was similar during both treatment periods, being 8404 kJ/day (2008 kcal/day) during the hydrolysed period and 8786 kJ/day (2100 kcal/day) during the control period. There were no statistically significant differences in the energy intake between the periods. Furthermore, the intake of carbohydrates, proteins, fats or fibres did not differ significantly between the two periods.

## 4. Discussion

Here, we provide further evidence that hydrolysed milk protein might be better tolerated than nonhydrolysed milk protein by people with functional gastrointestinal disorders. Both symptom sum scores (IBS-SSS and daily sum score) displayed improvements during the hydrolysed casein lactose-free period when compared to the intact milk protein lactose-free period (control). However, the detected difference in the effects of the products could not be explained by the biochemical measurements that we had hypothesised might reflect putative physiological imbalances in subjects with FGDs, i.e., the presence of low-grade inflammation, increased intestinal permeability or immune activation [1].

The magnitude of the effect was 38 points out of the baseline score of 199 on the IBS-SSS scale, which translates to a 19% reduction of the symptoms; the other symptom assessment score, a 10-day average of daily symptom recordings, was in-line with the IBS-SSS outcomes, giving further evidence that it was a true effect. However, these reductions in the symptoms must be interpreted as moderate, or even modest, because, quite commonly, a reduction of at least 50 points in the IBS-SSS is referred to as clinically meaningful [17].

Preliminary human data indicates that milk protein may cause symptoms in sensitive or inclined individuals without lactose intolerance or milk allergy [8,9,10]. To the best of our knowledge, only one previous study has evaluated the specific effects of casein-hydrolysed milk products [11]. In that study, in agreement with our present results, casein-hydrolysed lactose-free milk was better tolerated than the regular lactose-free milk tested [11]. These findings, together with our current results, suggest that milk protein, especially casein, may play a role in triggering gastrointestinal symptoms, at least in some individuals with self-reported milk sensitivities, even in the absence of lactose intolerance. On the other hand, milk homogenisation, another often-mentioned possible symptom trigger, does not seem to play a role in milk sensitivity, according to several randomised studies [18,19,20,21,22].

During recent years, the A1 genetic variant of β-casein has been associated with increased gastrointestinal symptoms, while A2 β-casein has been associated with reduced individual symptoms in healthy subjects; i.e., some, but not in all, randomised studies indicate that subjects may have a better tolerance towards A2 β-casein [23,24]. In our enzymatic hydrolysation method, the majority of β-casein (75%) is hydrolysed, and both A1 and A2 β-casein are similarly affected, whereas whey proteins remain intact. Thus, it is possible that the reduction either in total β-casein or only in A1 β-casein during hydrolysation could explain the reduction in gastrointestinal symptoms.

Collectively, these data suggest that intact casein in milk, rather than the homogenisation of milk, might be one possible culprit behind milk-related gastrointestinal symptoms when the role of lactose is ruled out, and thus, reducing the casein content of milk products might improve their tolerability in people affected by FGDs.

The key mechanistic features of FGDs include visceral hypersensitivity, dysbiotic changes in the microbiota, immune activation and increased intestinal permeability [1]. In our study, we did not find any statistically significant differences in the measures of the surrogate markers (IL-6, TNF-α, FABP2 and 1-methylhistamine) of these physiological attributes. The IL-6 concentration was elevated versus the baseline values during the hydrolysed casein period, which was against our hypothesis, but no change was detected in the concentration of TNF-α. The post hoc qualitative assessment of the patient diaries showed that two people had a relatively severe colds during the period when they were consuming the hydrolysed product, which might have theoretically affected their IL-6 values, where none had a cold during the control period. However, it remains impossible to determine if the episodes of colds truly skewed our IL-6 results. Another marker of immune activation, the level of 1-methylhistamine, a metabolite of histamine, was reduced in urine during the hydrolysed period when compared to the baseline but not when compared to the levels measured during the regular milk protein period. These seemingly contradictory findings open up the possibility that histamine may be a more sensitive marker of immune activation than classical blood markers of low-grade inflammation in cases of functional gastrointestinal disorders; naturally, this hypothesis must be tested in future randomised studies; our data can only act as a starting point for these investigations.

Interestingly, previous systematic reviews have suggested that an over-representation of mast cells, which secrete histamine in gut epithelia, is the most common and prominent marker of increased immune activation in IBS [25,26]. Furthermore, in a previous study, a low-FODMAP diet reduced urine histamine levels by approximately 80% and effectively reduced symptoms when compared to a high-FODMAP diet [27]. Accordingly, the treatment with an antihistamine ebastine also reduced IBS symptoms in a placebo-controlled trial [28]. Summing up, it might be valuable to determine if a dysfunctional histamine metabolism in relation to the diet plays a role in IBS and other functional gastrointestinal disorders.

Our study has several strengths; it was a double-blind, crossover intervention trial, which is rather rarely possible in the realm of dietary studies [29]. We used a validated symptom measurement, IBS-SSS, and adherence to the study products was excellent. Study subjects also recorded daily symptoms at the end of each day; both the IBS-SSS and the daily symptom score showed consistent improvements in symptoms during the period when they drank the hydrolysed milkshake. The trend of individual symptoms was also in-line with the total symptoms scores, albeit not statistically significant. Patients refrained from other dairy products during the interventions, and we also monitored possible dietary changes but found no differences in macronutrient intakes between the study periods. On the other hand, a ten-day treatment period may be too short to elicit effects on inflammatory markers, and the symptom results must be confirmed in longer prospective studies. We used a high protein dose (50 g/day), and it is uncertain if the results could be attained with a smaller amount of protein. Interestingly, one previous study detected differences in symptoms between intact and hydrolysed milk proteins at a dose as low as 7 g/d [11]. Lastly, our study population consisted of IBS, dyspepsia, diarrhoea and bloating patients, and this mixture of different functional gastrointestinal disorders may make interpretations of the results more complex.

## 5. Conclusions

In conclusion, a hydrolysed milk product with a lower casein content reduced gastrointestinal symptoms in sensitive individuals when compared to an intact milk protein product, but the mechanism behind the improved tolerance remains to be explained. The modest reduction in symptoms, despite the high protein dose, implies that casein explains only a small part of the food-related gastrointestinal symptoms in IBS and FGDs. This was also reflected by the reduction in the overall symptom score being 19% but less than 50 points in the IBS-SSS.

Nevertheless, lowering the casein content by partial hydrolysation could improve the tolerability to dairy products in individuals who suffer from FGDs.

## Figures and Tables

**Figure 1 nutrients-12-02140-f001:**
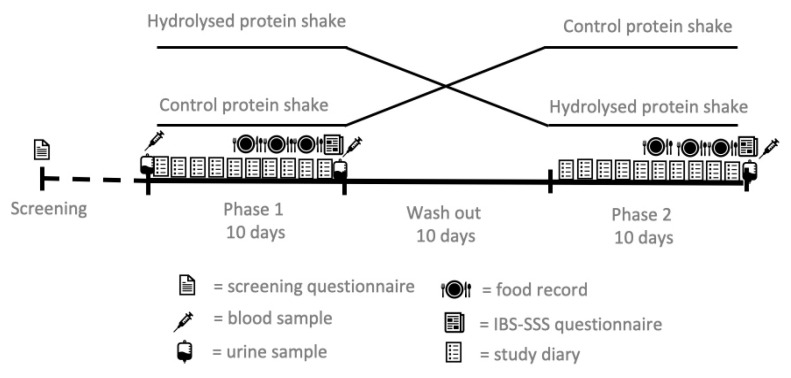
Study design.

**Figure 2 nutrients-12-02140-f002:**
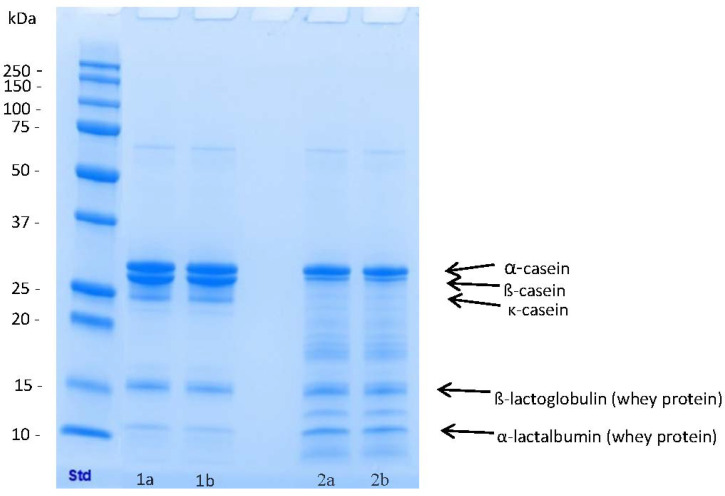
Sodium dodecyl sulphate-polyacrylamide gel electrophoresis (SDS-PAGE) of the study products. Std (Bio-Rad 161-0373), (1a and 1b) control milkshake and (2a and 2b) hydrolysed milkshake. Major milk proteins are identified.

**Table 1 nutrients-12-02140-t001:** Baseline characteristics of the subjects (mean ± SD). BMI: body mass index. IBS: irritable bowel syndrome.

	All *n* = 41	Women *n* = 33	Men *n* = 8
Age, years	44.0 ± 10.8	44.9 ± 10.9	40.6 ± 10.2
Height, cm	170.0 ± 10.8	166.8 ± 5.8	182.3 ± 8.3
Weight, kg	75.8 ± 16.2	72.2 ± 15.2	90.2 ± 11.8
BMI, kg/m^2^	26.2 ± 4.9	26.0 ± 5.3	27.1 ± 2.9
Diagnosis			
IBS	23	17	6
Functional dyspepsia	3	3	0
Functional diarrhoea	6	5	1
Functional bloating	9	8	1

**Table 2 nutrients-12-02140-t002:** Nutritional composition of milkshakes (per 100g of product).

	Control	Hydrolysed
Energy, kJ/kcal	335/80	322/77
Protein, g	10.6	10.3
Fat, g	1.6	1.5
Carbohydrates, g	5.7	5.5
Lactose, g	<0.01	<0.01

**Table 3 nutrients-12-02140-t003:** Scores of the irritable bowel syndrome-symptom severity scale (IBS-SSS) questionnaire and daily reported gastrointestinal symptoms (mean ± SD; *n* = 37).

	Baseline	Hydrolysed	Control	*p*-Value Hydrolysed versus Control ^†^
IBS-SSS	199 ± 71	163 ± 74 *	205 ± 75	0.001
Symptom score	14.7 ± 3.9	12.5 ± 2.4 *	14.0 ± 3.4	0.002
Abdominal pain	1.89 ± 0.74	1.61 ± 0.56	1.85 ± 0.67	0.09
Bloating	2.46 ± 1.00	2.05 ± 0.70 *	2.17± 0.88 *	0.72
Flatulence	2.49 ± 0.77	2.12 ± 0.61 *	2.32 ± 0.63	0.16
Rumbling	1.68 ± 0.81	1.39 ± 0.48	1.51 ± 0.59	0.25
Diarrhoea	1.85 ± 0.82	1.55 ± 0.64	1.62 ± 0.68	0.89
Constipation	1.45 ± 0.70	1.42 ± 0.54	1.54 ± 0.74	0.74
Heartburn	1.31 ± 0.56	1.09 ± 0.20 *	1.25 ± 0.40	0.01
Rapid feeling of fullness	1.55 ± 0.97	1.40 ± 0.59	1.38 ± 0.70	0.37

^†^ Repeated measures ANOVA for crossover design. * Significant difference from the baseline (*p* < 0.05).

**Table 4 nutrients-12-02140-t004:** Changes in the concentrations of the markers of inflammation, gut permeability and immune activation during the study (mean ± SD; *n* = 37). IL-6: interleukin 6. TNF-α: tumor necrosis factor alpha. FABP2: fatty acid binding protein 2

	Baseline	Hydrolysed	Control	*p*-Value ^†^
Plasma IL-6, pg/mL	1.16 ± 0.94	1.60 ± 1.34 *	1.28 ± 1.23	0.20
Plasma TNF-α, pg/mL	0.82 ± 0.35	0.84 ± 0.37	0.84 ± 0.38	0.48
Plasma FABP2, ng/mL	1.29 ± 0.47	1.42 ± 0.68	1.26 ± 0.52	0.13
Histamine ^‡^	131 ± 41	111 ± 37 *	117 ± 44	0.43

^†^ Repeated measures ANOVA for the crossover design. * Significant difference from the baseline (*p* < 0.05). ^‡^ Analysed as urinary 1-methylhistamine, ng/mg creatinine.

**Table 5 nutrients-12-02140-t005:** Dietary intakes of macronutrients and fibres during the treatment periods (mean ± SD).

	Hydrolysed	Control	*p*-Value ^†^
Energy, KJ/day	8404 ± 1753	8786 ± 2352	0.139
Energy, Kcal/day	2008 ± 419	2100 ± 562	0.139
Protein, g/day	121 ± 28	117 ± 30	0.562
Fat, g/day	76 ± 23	79 ± 33	0.442
Carbohydrates, g/day	185 ± 50	198 ± 65	0.111
Fibre, g/day	27 ± 9	27 ± 9	0.872

^†^ Comparison between study periods assessed by a paired *t*-test.

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
