# Peer review of "Randomised Controlled Trial: Partial Hydrolysation of Casein Protein in Milk Decreases Gastrointestinal Symptoms in Subjects with Functional Gastrointestinal Disorders"

_nutrients, 2020, doi:10.3390/nu12072140_

Round 1

Reviewer 1 Report

This is an interesting small sized cross-over placebo controlled trial in 41 adults with functinal gastrointestinal disorders.

They were given 500 ml/day of a chocolate flavored milkshake containing a high amount of protein ( 50g) either intact protein or partially hidrolised protein.  with the administration of the partially hydrolised milk product  a signficant redcution in IBS symptom score decreased. Heartburn was less reported compared to controls. Comapred to baseline, both milkshakes induced a reduction in bloating but only with the hydrolised one, heartburn and flatulence decreased .

The authors studied inflamatory, immune activation and intestinal permeability markers in order to unravel the mechanism of this reduction in symptoms, but no differences were found.

The manuscript is well written, original, and I have no major concerns to disclose. I have only several comments that should be adressed by the authors.

First: participants are 33 women  and 8 men. Do you think that this gender unbalance impairs generalization of results?

Second: The functional disorders included are rather heterogeneous. In my opinion, IBS patients could be analysed in the same group than functional diarrhea or functional bloating patients, but i do no think that patients with  functional dispepsia should be included in the same group. The authors should explain the reason for this inclusion

Third: 50 grams of protein is a huge amount of protein even for adults.  The observed worsening in IBS SSS in the control group might even be related to that intake of protein. This point should be clearly explained by the authors

Fourth: It surprises me the inflamatory markers were assesed in blood, but not in stools. Calprotectin determination is a sensitive marker of inflammation and easier to obtain than a blood sample. Why this marker was not included in the study?

Author Response

Responses to reviewer 1

We would like to thank the reviewer for the valuable comments and relevant questions, and for the possibility to respond to them.

Comment:

First: participants are 33 women and 8 men. Do you think that this gender unbalance impairs generalization of results?

Response:

This is a relevant question, but we consider this a common split of participants in clinical IBS studies. Women have approximately 2-4 times higher prevalence of IBS; so we think that our patient population captures the true nature of IBS quite well.

Comment:

Second: The functional disorders included are rather heterogeneous. In my opinion, IBS patients could be analysed in the same group than functional diarrhea or functional bloating patients, but i do no think that patients with functional dyspepsia should be included in the same group. The authors should explain the reason for this inclusion.

Response:

As we planned the study we thought about this issue a lot. We even consulted a globally renowned IBS gastroenterologist for his opinion.

We have two major reasons not to include solely IBS patients. First, we had no previous data on which patient populations would respond and which would not. We thought that it may well be bloating or IBS patients that would possibly react –but could not exclude functional dyspepsia or diarrhea patients either as there is some data showing that quite unexpectedly patients with dyspepsia may react to substances that mainly cause symptoms via the colon, i.e. FODMAPs (Masey et al. 2018). Also, our table 3 shows that it was only upper GI symptom (heartburn) that changed significantly during the treatments. This supports the inclusion of also patients with sole upper GI symptoms.

Secondly, the Rome IV criteria based prevalence of IBS is much smaller and pain driven when compared to the previous Rome criteria (about 30-40 % of Rome III prevalence, Sperber et al. 2020 https://doi.org/10.1053/j.gastro.2020.04.014 ). We wanted to have a broad enough patient group to test if milk protein is related to any specific symptom pronouncedly.

Taken together, we did not want to have purely pain-dominated IBS patients as we had no previous knowledge which particular symptom might drive milk protein related functional GI symptoms. Nonetheless, we agree that the mix of the patient population may make interpretation more complex, which we have now emphasized more clearly in the discussion.

Comment:

Third: 50 grams of protein is a huge amount of protein even for adults. The observed worsening in IBS SSS in the control group might even be related to that intake of protein. This point should be clearly explained by the authors.

Response:

We agree that the daily dose of milk protein was high and could have caused symptoms in our subjects. However, as this was a cross-over trial, all subjects consumed both products and served as their own controls. The intake of milk protein was the same (50 g) also when the control milk shake was consumed and as our results show, no significant differences in symptoms were observed between baseline and the control period.

We mention in the discussion that the dose of milk protein is high and it is possible that similar results may not have been achieved with a smaller dose in the short duration of the study.

Comment:

Fourth: It surprises me the inflammatory markers were assessed in blood, but not in stools. Calprotectin determination is a sensitive marker of inflammation and easier to obtain than a blood sample. Why this marker was not included in the study?

Response:

Fecal calprotectin is indeed a relevant and powerful marker for overt gut inflammation. It is used for screening and differential diagnosis of Crohn’s disease, UC and colorectal cancer.

However, it is broadly accepted that it is not useful in IBS or other functional gastrointestinal disorders, as gut inflammation is classified as minor “low grade” inflammation in functional GI disorders –not overt as in inflammatory bowel diseases.

https://www.hindawi.com/journals/grp/2015/490183/ 

Therefore, calprotectin is not used for IBS. Most clinical studies until today have therefore used blood markers of low-grade inflammation /immune activation rather than calprotectin. For these reasons, we also used blood markers. Lipopolysaccharide (LPS) is a fecal marker that could have been used, but we chose blood markers this time.

Reviewer 2 Report

The authors conducted a well-designed randomized, cross-over trial statistically designed to detect a 50-point difference between study projects according to the IBS-SSS 500 point scale. The study findings are of interest, but the presentation of results and conclusions diverge from what was originally stated as the primary outcome. Therefore, a number of modifications are required. 

Major comments:

  1. The study was designed based on the ability to detect a 50-point difference between study products on the IBS-SSS score. This should be treated as the primary outcome. All other outcomes are secondary (e.g. outcomes listed on page 6, lines 191-198) and should be presented in the results section as such. Furthermore, the results of the primary outcome specifically need to be articulated in the results section
  2. Recommend providing an additional breakdown of the IBS-SSS category scores. Given the reported symptoms of abdominal pain, bloating, and flatulence between products provided in Table 3, it's not clear what is driving the score to be even higher during control than baseline.
  3. Dietary intake results do not appear to have been included with the supplemental material. Regardless, this data needs to be presented in the main manuscript. This is an enormous potential confounding variable that should be given more weight when interpreting symptom differences between interventions. Although differences reportedly failed to reach statistical significance that does not preclude clinical significance.
  4. The conclusion needs to be adjusted to account for the failure to achieve the primary outcome (e.g. 50 point difference between interventions)

Author Response

Responses to reviewer 2

We would like to thank the reviewer for the valuable comments and relevant questions, and for the possibility to respond to them.

Comment:

The study was designed based on the ability to detect a 50-point difference between study products on the IBS-SSS score. This should be treated as the primary outcome. All other outcomes are secondary (e.g. outcomes listed on page 6, lines 191-198) and should be presented in the results section as such. Furthermore, the results of the primary outcome specifically need to be articulated in the results section.

Response:

We realize that the primary outcomes were not defined unambiguously enough in the original text. The power calculation was based on IBS-SSS, and we have now clarified the text by separating the power calculation from the primary outcomes in the Statistical analyses section.

Comment:

Recommend providing an additional breakdown of the IBS-SSS category scores. Given the reported symptoms of abdominal pain, bloating, and flatulence between products provided in Table 3, it's not clear what is driving the score to be even higher during control than baseline.

Response:

Thank you for the comment.

We used two symptoms scores: the IBS-SSS and VAS scores. None of the symptom scores are statistically significantly different between baseline and control as stated in the text. The IBS-SSS also decreased significantly from baseline during the period with the hydrolysed product (p=0.001), but not during the period when they drank the control milkshake (p=0.68).

As one observes, the daily reported VAS scores (table 3) show that none of the individual scores drive the change predominantly.

Most symptoms show a tendency towards being higher on regular milk protein, albeit for only heartburn, the difference between the treatments is statistically significant. The trends can be seen more clearly in figures on the development of individual daily reported symptoms during the ten-day study periods, which we have added in the supplemental material ( Figure S3 (a-1): The development of total symptom score and individual symptom scores during the study periods).

Comment:

Dietary intake results do not appear to have been included with the supplemental material. Regardless, this data needs to be presented in the main manuscript. This is an enormous potential confounding variable that should be given more weight when interpreting symptom differences between interventions. Although differences reportedly failed to reach statistical significance that does not preclude clinical significance.

Response:

We sincerely apologize for mistakenly not including the table on dietary intakes in the supplemental material, as was intended. We have now included this table in the main manuscript (Table 5, page 8) and are happy to report that no differences between the treatment periods were demonstrated in the dietary intakes.

Comment:

The conclusion needs to be adjusted to account for the failure to achieve the primary outcome (e.g. 50 point difference between interventions).

Response:

We agree that the reduction in symptoms was modest, despite the high protein dose. We have now stated more clearly in the conclusions that the reduction in IBS-SSS was less than 50 points:

“The modest reduction in symptoms, despite the high protein dose, implies that casein explains only a small part of food-related gastrointestinal symptoms in IBS and FGDs. This was also reflected by the reduction in overall symptom score being 19%, but less than 50 points in IBS-SSS.”